# Deep Learning-Based Automated Magnetic Resonance Image Segmentation of the Lumbar Structure and Its Adjacent Structures at the L4/5 Level

**DOI:** 10.3390/bioengineering10080963

**Published:** 2023-08-15

**Authors:** Min Wang, Zhihai Su, Zheng Liu, Tao Chen, Zhifei Cui, Shaolin Li, Shumao Pang, Hai Lu

**Affiliations:** 1Department of Spinal Surgery, Fifth Affiliated Hospital of Sun Yat-Sen University, 52 Meihua Dong Lu, Xiangzhou District, Zhuhai 519000, China; 19002098550@163.com (M.W.); 13265092954@163.com (Z.S.); spineliuzheng@163.com (Z.L.); eagle121@163.com (T.C.); 15626452678@163.com (Z.C.); 2Department of Radiology, Fifth Affiliated Hospital of Sun Yat-Sen University, 52 Meihua Dong Lu, Xiangzhou District, Zhuhai 519000, China; lishlin5@mail.sysu.edu.cn; 3School of Biomedical Engineering, Guangzhou Medical University, No. 1, Xinzao Road, Xinzao Town, Panyu, Guangzhou 511436, China

**Keywords:** magnetic resonance imaging, neural network models, deep learning, preoperative plan, 3D visualization

## Abstract

(1) Background: This study aims to develop a deep learning model based on a 3D Deeplab V3+ network to automatically segment multiple structures from magnetic resonance (MR) images at the L4/5 level. (2) Methods: After data preprocessing, the modified 3D Deeplab V3+ network of the deep learning model was used for the automatic segmentation of multiple structures from MR images at the L4/5 level. We performed five-fold cross-validation to evaluate the performance of the deep learning model. Subsequently, the Dice Similarity Coefficient (DSC), precision, and recall were also used to assess the deep learning model’s performance. Pearson’s correlation coefficient analysis and the Wilcoxon signed-rank test were employed to compare the morphometric measurements of 3D reconstruction models generated by manual and automatic segmentation. (3) Results: The deep learning model obtained an overall average DSC of 0.886, an average precision of 0.899, and an average recall of 0.881 on the test sets. Furthermore, all morphometry-related measurements of 3D reconstruction models revealed no significant difference between ground truth and automatic segmentation. Strong linear relationships and correlations were also obtained in the morphometry-related measurements of 3D reconstruction models between ground truth and automated segmentation. (4) Conclusions: We found it feasible to perform automated segmentation of multiple structures from MR images, which would facilitate lumbar surgical evaluation by establishing 3D reconstruction models at the L4/5 level.

## 1. Introduction

Over the past 30 years, low back pain has been one of the leading causes of disability worldwide, burdening individuals, healthcare, and society [1,2,3]. Lumbar magnetic resonance imaging (MRI) has evolved into an essential non-invasive diagnostic tool for detecting and preoperative assessment of patients with low back pain [4,5]. More critically, for some minimally invasive surgical (MIS) procedures, such as the MIS oblique approach, preoperative MR image analysis can help to determine whether there is a sufficient surgical window [6]. It can also help to determine the anatomical variation of the large blood vessels in the surgical area. This preoperative evaluation, a common practice in the MIS oblique approach, aims to ensure the procedure’s success and estimate the risk of postoperative complications. Several complications, such as large blood vessel damage, nerve injury, and psoas muscle injury, however, may occur in cases of inadequate preoperative evaluation [7].

As a result of the complexity of the three-dimensional (3D) anatomy of the lumbar spine and the sensitive nature of the neurovascular structures involved, the limited information available in two-dimensional (2D) images poses a challenge for preoperative and operative evaluation [8,9]. The space structures of 3D surgical views are often unobserved directly by radiologists or surgeons due to the limited structural information on a single CT or MR slice [10]. Some researchers have attempted to develop a new preoperative evaluation method to acquire 3D geometric morphology using multi-planar MR images for MIS procedures [11]. Even though the 3D information was obtained before or during the surgery through multi-planar MR images, including sagittal, coronal, and axial images, the inconvenient truth is that the surgeons’ minds were not effectively able to interpret these complicated 3D anatomical structures in clinical practice. 

The improvements in perioperative patient safety for the MIS oblique approach come from knowledge of 3D anatomy. However, manual or semi-automatic segmentation of the lumbar structures and their adjacent structures in clinical practice is laborious and time-consuming. It also suffers from intra- and inter-reader variability [12]. Deep learning, a machine learning technique that employs multi-layer neural networks, has been extensively utilized for automatic medical image segmentation [13,14]. To our knowledge, however, only a handful of studies have simultaneously segmented lumbar structures and their adjacent structures on MR images simultaneously [15,16]. Prior studies and segmentation have focused mainly on bones [17,18], discs [19], and nerves [20]. Regrettably, the segmentation of automatic lumbar structures and their adjacent structures in MR images has not been systematically investigated before, specifically for 3D segmentation of the large blood vessels, as well as the psoas major muscle. Therefore, 3D segmentation of lumbar structures and their adjacent structures using deep learning is a pressing and unmet clinical need.

The purpose of this study was to develop a deep learning model using a modified 3D Deeplabv3+ network to automatically segment lumbar structures and their adjacent structures (including bones, intervertebral disc, nerve roots, dura, abdominal aorta, inferior vena cava, and psoas major) from MR images at the L4/5 level. Specifically, we compared the performance of the deep learning model to that of the performance of manual segmentation through the evaluation of quantitative metrics (including DSC, precision, and recall) and the morphometric-related measurements of the 3D lumbar model generated by image segmentation. If successful, the deep learning model could be used for automated segmenting of lumbar structures and their adjacent structures, which would allow for the preoperative and operative evaluation of the MIS oblique approach in clinical practice to improve perioperative patient safety.

## 2. Materials and Methods

### 2.1. Study Subjects

This study was approved by the institutional ethics committee of the Fifth Affiliated Hospital of Sun Yat-sen University (NCT04647279, IRB-2020 K05-1) and was conducted in accordance with the Helsinki Declaration of 1975, as revised in 2013. All participants signed informed consent forms. The study recruited a total of 50 participants, encompassing individuals diagnosed with lumbar degenerative diseases based on clinical predictions as well as healthy volunteers. The inclusion criteria required participants to be at least 18 years old and have no contraindications for MRI examination. Exclusion criteria included individuals with a history of spinal surgery, subjects with spinal deformity, and patients diagnosed with lumbar spondylolisthesis. A total of 50 participants who had undergone a 3T MRI with T2-3D-space sequences (TR/TE, 2800.0/189.0; FA, 45°; FOV, 240 × 240 mm; Matrix, 320 × 320; Slice thickness, 0.8 mm; Bandwidth, 579 kHz; and Resolution, 0.8 × 0.8 × 0.8 mm.) at the L4/5 level between March and July 2020 were recruited in the present study. Of the 50 L4/5 levels, 41 were healthy and 9 unhealthy, with spinal stenosis at 1 level, disc herniation at 8 levels, or both at 1 level. L4/L5 data at the lumbar level and the demographic characteristics of the participants were recorded (Table 1). Performance of the automatic segmentation model was evaluated in the current study. A 3D model of multiple structures in the L4/5 level was constructed using manual and automatic segmentation images. All key parameters of the model were recorded.

### 2.2. Image Annotation and Preprocessing

Trained professionals manually segmented all image data using Mimics software (Mimics^®^, 3-matic, Materialise, Inc., Leuven, Belgium). Manually annotated anatomical structures included the L4 vertebrae, L5 vertebrae, the abdominal aorta, the inferior vena cava, psoas major muscles, and intervertebral discs. In addition, all the segmented images were evaluated by a radiologist and a spinal surgeon. The three doctors had an open discussion to confirm the accuracy of the segmentation data in case of a dispute.

Images were further subjected to cropping, normalization, and padding preprocessing steps. Given an image with specifications of $I\in R^{D\times H\times W}$, the cropped version of an image was obtained using the expression presented below:$$I_{crop}(I)=I\left[:,\frac{1}{4}H-40:\frac{3}{4}H+40,\frac{1}{4}W-10:\frac{3}{4}W+10\right]$$
where D, *H*, and *W* represent the depth, height, and width of the image, respectively. The value of D in the current study varied from 88 to 128. Cropped image size was D×240×180; the voxel values were then normalized by subtracting the average and dividing by the standard deviations. The normalized images were ultimately zero-padded to attain a size of 128×240×180.

### 2.3. Model Architecture

A 3D DeepLabv3+, modified from previous reports [15,21], was utilized to attain the automatic segmentation of multiple anatomical structures in L4/5 segment MR images (Figure 1). The 3D DeepLabv3+ comprises a deep convNet (Figure 2) for extracting low-level (stride = 2) and high-level image representation (stride = 4). In addition, it has a decoder for generating the predicted segmentation. Moreover, the output of the decoder contains 9 channels including 8 anatomical structures and the background.

### 2.4. Experimental Configurations

A 5-fold cross-validation method was utilized to evaluate the performance of the model. The 50 subjects were randomly assigned to 5 groups, each with 10 individuals. Ten subjects from a group were used as the test dataset in each experiment, whereas thirty-two individuals, randomly selected from the other 4 groups, were used as the training dataset. The other 8 subjects were used as the validation dataset. To enhance generalization, the training dataset was augmented online with random rotations ranging from −15° to 15° and random elastic deformations. The 3D DeepLabv3+ was trained for 200 epochs using the training dataset. The validation dataset was used to calculate the Dice Similarity Coefficient after training at each epoch, and the trained model with the highest Dice Similarity Coefficient among the 200 epochs was used for testing. Five replicates were conducted for this process until the segmentation results of all the samples were obtained. Moreover, the DeepLabv3+ model, developed by Facebook Artificial Intelligence Research and implemented in PyTorch version 1.5.1, underwent training for 200 epochs using the Adam optimizer with a batch size of 2. The learning rate was initially set at 0.005 and then lowered 5 times at epochs 66 and 133. 

### 2.5. Examination of Model Performance and Morphometric Evaluation 

The Dice Similarity Coefficient (DSC) [22] was utilized as a quantitative metric to assess the segmentation performance of DeepLabv3+. Notably, all metrics were calculated for a single object in the original image space (including L4, L5, and abdominal aorta, etc.). In addition, the average of the DSC of all the objects was calculated to obtain the final result. Two independent researchers evaluated segmentation effect of DeepLabv3+ in clinical application. Mimics were initially used to establish 3D reconstruction of the field of view of the L4/5 level. Moreover, the 3D models were cut along the middle section of the mask of the L4 vertebral body and the intervertebral disc. Morphological data related to the surgery in the field of view of the L4/5 level were then obtained (Figure 3). One of the observers re-evaluated all the morphological parameters of the three-dimensional model of multiple structures in the L4/5 level a month after the first analysis. The final morphological parameters were obtained by averaging the values of the three measurements. 

### 2.6. Statistical Analysis

Statistical tests were conducted using SPSS 25.0 (IBM Corporation, Chicago, IL, USA). Differences in morphometric parameters of the three-dimensional model of multiple structures in L4/5 level, between automatic and manual segmentation were evaluated using the Wilcoxon signed-rank test without assuming the underlying distribution. A *p*-value < 0.05 was considered statistically significant. Pearson correlation coefficient, Bland–Altman plot, and scatter diagram were used to evaluate the reliability of the morphological analysis. 

## 3. Results

### 3.1. Performance of Automatic Segmentation

The validation results showed that DeepLabv3+ was effective in the segmentation of the spinal structures (L4 vertebrae (L4), L5 vertebrae (L5), Intervertebral disc (IVD), L4 nerve roots (N4), dura, Abdominal Aorta (AA), Inferior Vena Cava (IVC), and Psoas major (PM)) on axial MR. The results of the three-dimensional visual model was established according to the results of manual (Figure 4A,C) and automatic (Figure 4B,D) segmentation of L4/5 MRI images were compared. Automatic segmentation showed a high DSC score (Table 2 and Figure 5). Analysis of the overall automatic segmentation effect of the test set revealed that 92% of the DSC of the model was above 0.85 and 15% was above 0.90. This indicated that the Deeplabv3+ model was effective in the 3D reconstruction of multiple structures in L4/5 on MRI.

The entire training of DeepLabv3+ lasted for about 12.5 h in each validation fold. The modified 3D Deeplabv3+ network took 3.5 s (a Quadro RTX 8000 GPU) per sample to complete an automated segmentation after training. This duration was significantly lower compared with the 300 min used in manual segmentation (Section 3.1).

### 3.2. Morphometric Correlation of Parameters and Difference between Manually and Automatically Segmented Images

Automatic and manual segmentation data were compared to establish a three-dimensional model of multiple structures in the L4/5 level (p, the Wilcoxon signed-rank test). The findings showed no significant differences between the automatic and manual measurements. The results showed a strong correlation between manual and automatic segmentation for establishing the 3D model of multiple structures in the L4/5 level (R, Pearson correlation coefficient). Details on the morphologic parameters are presented in Table 3. The D value of each group corresponded to the mean scatter plot (Bland–Altman plot) generated. Notably, the points representing fluctuations in the D-value were close to the average of the D-value, with almost all D-value points located within the 95% consistency interval. Moreover, each group of points in the scatter plot was located near the tropic line (Figure 6 and Figure 7). The results indicated that the 3D-reconstructed model of multiple structures at the L4/5 level, established by automatic segmentation, exhibited high reproducibility of the measurements from the manual segmentation model.

## 4. Discussion

Utilizing deep learning-based image segmentation has the potential to accelerate the 3D reconstruction of multiple structures at the L4/5 level. The modified 3D Deeplabv3+ network [21] was well demonstrated in this study due to its multi-scale and multi-structural segmentation advantages, which took 3.5 s (a Quadro RTX 8000 GPU) per sample to complete an automated segmentation after training. This duration was significantly lower compared with the 300 min used in manual segmentation. In this study, five-fold cross-validation was employed to examine an automated MRI segmentation technique based on deep neural networks. According to the findings, deep learning techniques can accurately and rapidly segment lumbar spinal structures (including bones, intervertebral discs, nerve roots, dura mater, abdominal aorta, inferior vena cava, and psoas major), thereby expediting the 3D reconstruction of multiple structures in the L4/5 region on magnetic resonance imaging. Our method shows promising potential for preoperative assessment of spatial safety in anterior surgery and comprehensive understanding of 3D anatomy within the surgical field.

The methods proposed in the present study showed good performance in automatic segmentation of the lumbar spine structure. The average DSC of the method developed in the current study for automatic segmentation of vertebral structure was 0.93. In addition, the segmentation performance was similar to that achieved using an automatic image segmentation model developed by Pang Shumao et al. (the average DSC was 0.94) [23]. The average DSC of automatic disc segmentation was 0.92, which was higher compared with the segmentation performance of the model developed by Pang Shumao et al. (the average DSC is 0.87). The present method achieved an average DSC of 0.90 and 0.72, respectively in automatic segmentation of the dura mater and nerve roots. Moreover, the total segmentation accuracy for the method was 0.81, which was similar to findings from our previous research (the average DSC of the dura mater and nerve roots was 0.84) [20]. In contrast to previous research, this study segmented multiple structures in the lumbar spine and its vicinity, including bones, intervertebral discs, nerve roots, dura mater, abdominal aorta, inferior vena cava, and psoas major. This provides a more detailed three-dimensional anatomical relationship that significantly enhances the clinical applicability of the segmentation model. The results of the automatic segmentation of the psoas major anatomical structure showed that the present model achieved an average DSC of 0.95, higher than the performance of the paravertebral muscle automatic segmentation model developed by Dourthe et al. (the average DSC of psoas major automatic segmentation was 0.90) [24]. Moreover, the model in the present study showed good performance in automatic segmentation of the abdominal aorta and inferior vena cava (the average DSC of abdominal aorta and inferior vena cava was 0.85 and 0.88, respectively). This performance was higher compared to the performance of the abdominal vascular automatic segmentation model developed by Golla’s research group (the average DSC of arteries was 0.84 and the average DSC of veins was 0.76) [25]. In addition, the intricate structure of nerve roots poses challenges in distinguishing them from other tissues. However, the overall segmentation performance of the model was satisfactory. These findings show that the automatic image segmentation method based on deep learning developed in the present study achieved the same or better segmentation performance as models developed in previous studies.

Accurate reconstruction of the 3D anatomy of the lumbar region can be used for preoperative 3D measurement and evaluation of some specific lumbar spine surgeries. Previous studies report that the distance between the psoas major muscle and the great vessels of the lumbar anterior spine (the safe zone distance for MIS oblique approach, OC, the oblique corridor) must be accurately assessed before surgery to avoid complications such as vascular injury [7]. Molinares et al. [6] reported that the average OC at the L4/5 level was 10.28 mm after carrying out MRI examination. Wang Hongli et al. [26] conducted an autopsy study and observed that the average distance between the anterior border of the left psoas muscle and the abdominal aorta was 8.90 mm at the L4/5 level. In this study, analysis using the 3D models generated by automatic segmentation and manual segmentation showed that the minimum of OC at the median level of the L4/5 intervertebral disc was 9.214 mm and 9.077 mm, respectively. Measurements between the two were similar to previously reported safe zone distances. In addition, the measurement parameters of the two models were determined and the differences were compared. The results showed that the measurements of the 3D model generated by automatic image segmentation were not statistically different from the measurements obtained from the 3D model generated by manual segmentation. This indicates that the automatic image segmentation performance of the method proposed in this study is similar to that of medical professionals. Currently, the three-dimensional visualization model of the lumbar spine plays a crucial role in preoperative planning and path trajectory evaluation for lumbar surgery [9,27]. In this study, we have meticulously elucidated the anatomical structure adjacent to the lumbar intervertebral foramen and anterior to the lumbar spine, which will greatly facilitate preoperative planning of minimally invasive procedures in the lumbar region (including the MIS oblique approach and intervertebral foramen endoscopy, among others). This may potentially reduce surgical complications and improve postoperative recovery outcomes.

The current study has some limitations. In the current study, automatic segmentation of the lumbar segmental artery was not explored. If the segmental artery can be automatically segmented and the three-dimensional spatial relationship between the segmental artery and lumbar region can be clearly presented, this will assist the evaluation and guidance of the MIS oblique approach, which will be more conducive to reducing the incidence of vascular injury complications. Further studies should be conducted to explore the automatic segmentation of lumbar segmental arteries and integrate it into the model constructed in the present study. Furthermore, the study cohort primarily consisted of younger individuals with relatively uncomplicated disease types, predominantly lumbar disc herniation and spinal stenosis patients. Therefore, we intend to expand our sample size to encompass diverse age groups, body types, and disease types (including spinal deformity and lumbar spondylolisthesis) to enhance the generalizability of our model. In addition, this study lacked clinical surgical validations. Further studies should be performed to conduct clinical validation of automatic segmentation of important tissue structures at the L4/5 level and explore the clinical feasibility of 3D visualization models.

## 5. Conclusions

In this study, a modified 3D Deeplabv3+ network-based deep learning model was developed to automatically segment multiple structures from MR images at the L4/5 level, achieving performance comparable to that of human experts. In addition, the deep learning model proved the ability to accurately reconstruct the 3D lumbar model with strong linear relationships and correlation for the morphometric-related measurements, comparing the 3D lumbar model of manual segmentation. We found it feasible to perform automated segmentation of multiple structures from MR images, which would facilitate minimally invasive lumbar surgical evaluation by establishing 3D reconstruction models at the L4/5 level.

## Figures and Tables

**Figure 1 bioengineering-10-00963-f001:**
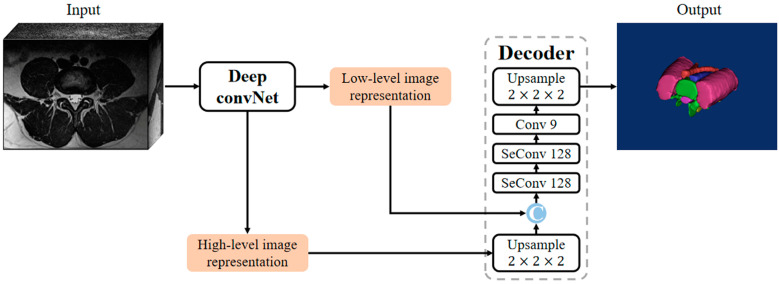
The framework of 3D DeepLabv3+, which is composed of a deep convNet and a decoder. SeConv # denotes a depthwise separable convolution with # output channels.

**Figure 2 bioengineering-10-00963-f002:**
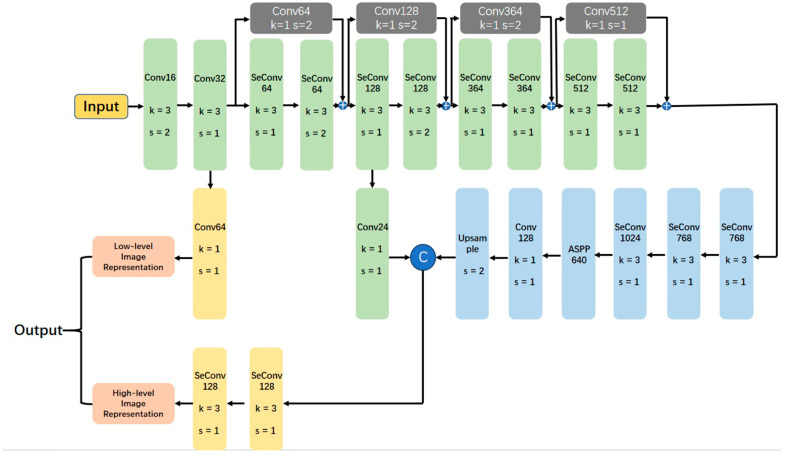
The framework of the deep convNet, which is used to generate low-level and high-level image representations. k=a and s=b show that the kernel size was a×a×a and the stride was b×b×b, respectively.

**Figure 3 bioengineering-10-00963-f003:**
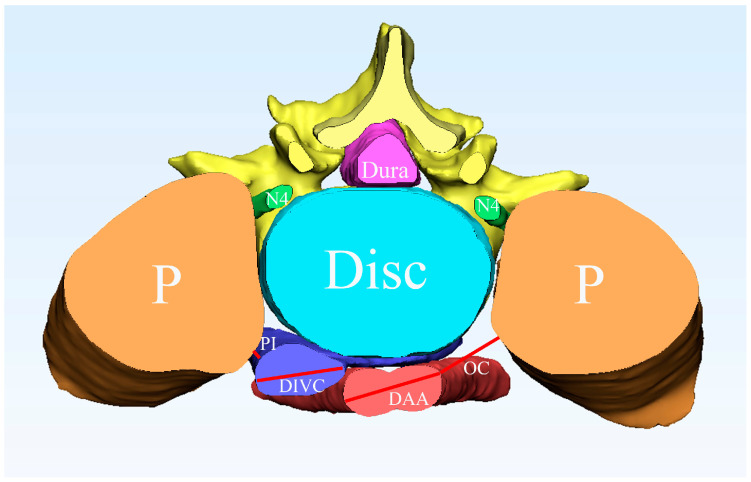
Morphological measurement using the three-dimensional model of the L4/5 level. P, psoas major; Disc, intervertebral disc; N4, Lumbar 4 nerve roots; OC, the shortest distance between the psoas major and abdominal aorta in the middle of the L4/5 intervertebral disc; DAA, the maximum diameter of the abdominal aorta in the middle of the L4/5 intervertebral disc; DIVC, the maximum diameter of the inferior vena cava in the middle of the L4/5 intervertebral disc; PI, the shortest distance between the psoas major muscle and the inferior vena cava in the middle of the L4/5 intervertebral disc.

**Figure 4 bioengineering-10-00963-f004:**
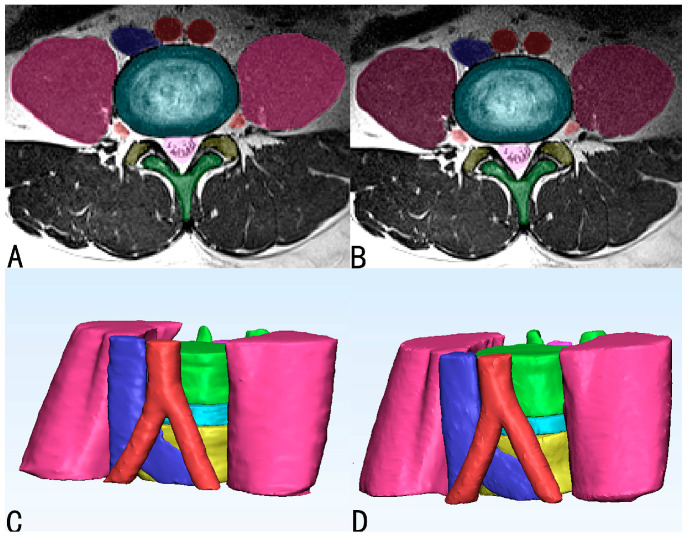
Results of manual and automatic segmentation of MRI images and the 3D model of L4/5 level. (**A**,**C**) Manual segmentation; (**B**,**D**) automatic segmentation.

**Figure 5 bioengineering-10-00963-f005:**
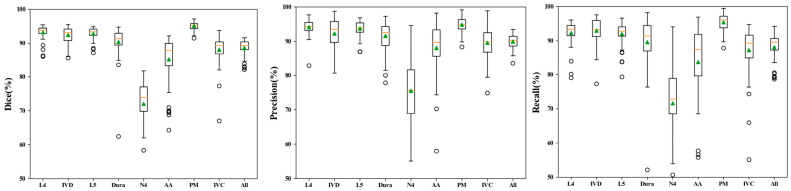
The 3D Deeplabv3+ achieved good performance in terms of DSC, precision, and recall for segmentations of various anatomical structures, including bones (L4 and L5), dura mater, discs, nerves, abdominal aorta, inferior vena cava, psoas major, and all 8 spinal structures at the L4/L5 level. The median value and mean value are represented by the orange line and green triangle in the box, respectively.

**Figure 6 bioengineering-10-00963-f006:**
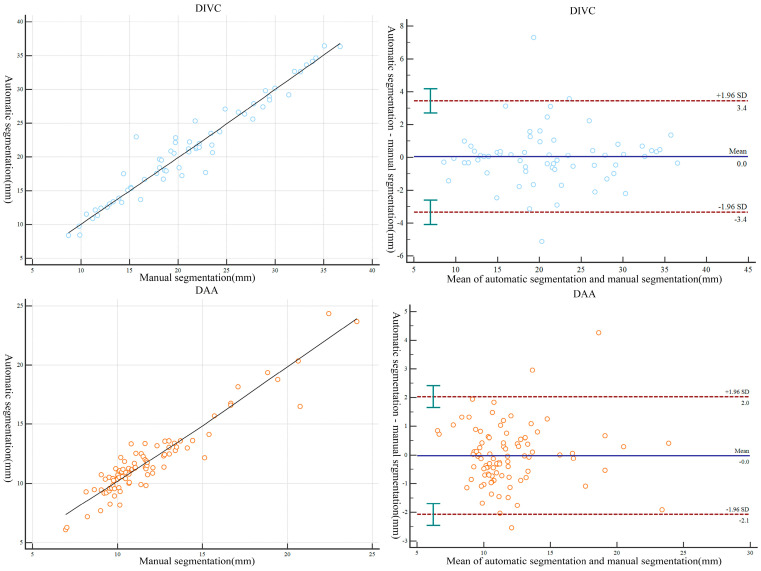
Scatterplots and Bland–Altman plots comparing measurements made by the L4/5 segment 3D model produced from manual and automatic segmentation methods. DAA, the maximum diameter of the abdominal aorta in the middle of the L4/5 intervertebral disc; DIVC, the maximum diameter of the inferior vena cava in the middle of the L4/5 intervertebral disc.

**Figure 7 bioengineering-10-00963-f007:**
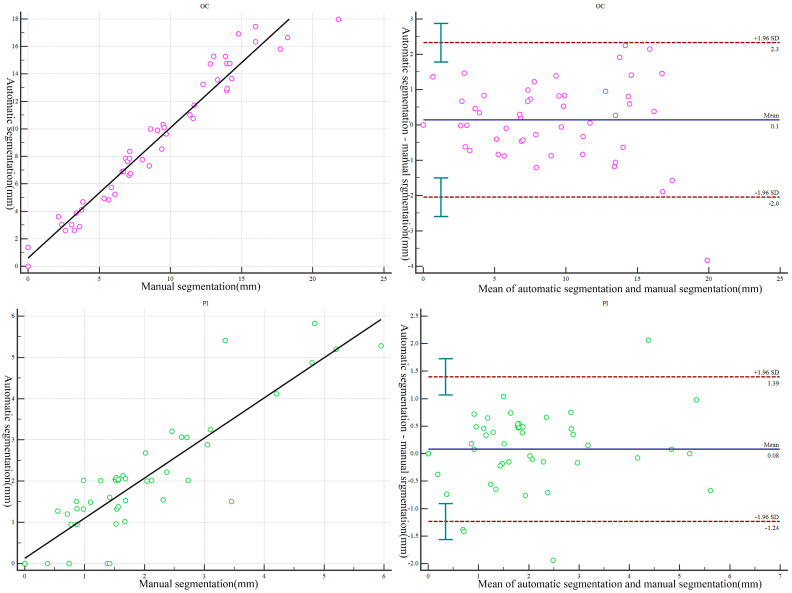
Scatterplots comparing measurements made by the L4/5 segment 3D model produced from manual and automatic segmentation methods. OC, the shortest distance between the psoas major and the abdominal aorta in the middle of the L4/5 intervertebral disc; PI, the shortest distance between the psoas major muscle and the inferior vena cava in the middle of the L4/5 intervertebral disc.

**Table 1 bioengineering-10-00963-t001:** Dataset demography.

Sequence and Parameters	L4/L5 Level Dataset
T2-3D-space	
Men ^II^	27 (54)
Women ^II^	23 (46)
Age ^III^	34.70 (23, 63)
Male Participants ^III^	32.37 (25, 46)
Female Participants ^III^	37.43 (23, 63)
Body Mass Index(kg/m^2^) ^IV^	23.79 (20.68, 26.90)

Note: T2-3D-space = T2-3D-weighted sampling perfection with application-optimized contrast with different flip-angle evolutions. ^II^ Data are numbers of participants, with percentages in parentheses. ^III^ Data are means, with ranges in parentheses. ^IV^ Data are means, with 95% confidence intervals in parentheses.

**Table 2 bioengineering-10-00963-t002:** Results of automatic segmentation performances.

Datasets		L4	IVD	L5	Dura	N4	AA	PM	IVC	All
DSC	Training	0.951	0.955	0.951	0.935	0.805	0.921	0.976	0.930	0.928
	Validation	0.926	0.921	0.925	0.895	0.700	0.860	0.951	0.878	0.882
	Test	0.931	0.924	0.928	0.903	0.721	0.852	0.950	0.880	0.886
Precision	Training	0.956	0.954	0.955	0.940	0.810	0.924	0.974	0.930	0.930
	Validation	0.938	0.925	0.930	0.910	0.731	0.894	0.952	0.904	0.898
	Test	0.941	0.923	0.937	0.915	0.756	0.881	0.948	0.895	0.899
Recall	Training	0.947	0.956	0.945	0.934	0.818	0.921	0.979	0.934	0.929
	Validation	0.916	0.920	0.921	0.886	0.703	0.835	0.951	0.862	0.874
	Test	0.923	0.928	0.920	0.896	0.717	0.837	0.953	0.872	0.881

Note: Data are means of Dice Similarity Coefficient of 5-Fold cross-validation scores. L4, L4 vertebral body; L5, L5 vertebral body; IVD, intervertebral disc. N4, L4 nerve roots; AA, abdominal aorta; IVC, inferior vena cava; PM, psoas major.

**Table 3 bioengineering-10-00963-t003:** Comparison of morphological measurement results between automatic and manual segmentation of 3D models “x ± s, (minimum–maximum)”.

Level	Morphometric Parameters	3D Model of Automatic Segmentation	3D Model of Manual Segmentation	Mean Absolute Error	R Value	*p* Value
	OC	9.214 ± 4.897 (0.000–17.970)	9.077 ± 5.038 (0.000–21.80)	0.866 ± 0.706 (0.000, 3.830)	0.975	0.259
	DAA	11.852 ± 3.150 (6.100–24.350)	11.828 ± 3.193 (6.950–24.10)	0.779 ± 0.697 (0.010, 4.260)	0.945	0.504
	DIVC	21.081 ± 7.257 (8.420–36.430)	21.039 ± 7.217 (8.710–36.680)	1.121 ± 1.312 (0.050, 7.300)	0.971	0.947
	PI	1.929 ± 1.504 (0.000–5.820)	1.852 ± 1.382 (0.000–5.950)	0.489 ± 0.462 (0.000, 2.060)	0.895	0.241

Note: Statistical significance is determined at the *p* < 0.05 level. OC, the shortest distance between the psoas major and abdominal aorta in the middle of the L4/5 intervertebral disc; DAA, the maximum diameter of the abdominal aorta in the middle of the L4/5 intervertebral disc; DIVC, the maximum diameter of the inferior vena cava in the middle of the L4/5 intervertebral disc; PI, the shortest distance between the psoas major muscle and the inferior vena cava in the middle of the L4/5 intervertebral disc.

## Data Availability

The dataset is not publicly available due to restrictions imposed by the data sharing agreements with the institutional ethics committee of the Fifth Affiliated Hospital of Sun Yat-sen University. However, a partial dataset can be obtained upon reasonable request to the corresponding authors for academic purposes.

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
