# Peer review of "Deep Learning-Based Automated Magnetic Resonance Image Segmentation of the Lumbar Structure and Its Adjacent Structures at the L4/5 Level"

_bioengineering, 2023, doi:10.3390/bioengineering10080963_

Round 1
Reviewer 1 Report
Introduction
Briefly define the term "deep learning" as you are using it.
Material & Methods
No specific details regarding patient consent. Your manuscript does not contain a complete IRB statement regarding ethics board approval. Original articles need to contain a statement about the Helsinki Declaration of 1975, as in the example given here: “This study was approved by the human subjects ethics board of XXXXX and was conducted in accordance with the Helsinki Declaration of 1975, as revised in 2013.
Study participants- the creation of the datasets should be better described- how were patients collected.
MRI protocol- For dataset please state what complete MRI sequence (TR, TE, FOV, etc) were used for the MRI protocol.
The conclusion focuses solely on the technical aspects of the deep learning model and its performance in segmenting structures from MR images. It would be beneficial to expand the discussion to include the potential clinical impact of these findings and how they could contribute to improving patient outcomes or medical decision-making.
Author Response
POINT-TO-POINT RESPONSE TO REVIEWERS
We would like to thank the reviewers and the Editors for their effort in evaluating this work and for their constructive comments and criticisms. The author team has discussed the review and carefully conducted a series of experiments to address the points raised. Several editorial changes were made in the main manuscript and those were indicated, as requested by the journal, and uploaded separately. The outcome of the review process has been a stronger study and therefore we are grateful.
Reviewer(s)' Comments to Author:
Reviewing: 1
Review Comments for Author(s)
Q1:
Introduction
Briefly define the term "deep learning" as you are using it.
Response:
We sincerely thank the reviewer for careful reading. We have made extensive corrections to our previous draft. The detailed corrections are listed as follows: “Deep learning, a machine learning technique that employs multi-layer neural networks, has been extensively utilized for automatic medical image segmentation.” (Line 70-71.)
Q2:
Material & Methods
No specific details regarding patient consent. Your manuscript does not contain a complete IRB statement regarding ethics board approval. Original articles need to contain a statement about the Helsinki Declaration of 1975, as in the example given here: “This study was approved by the human subjects ethics board of XXXXX and was conducted in accordance with the Helsinki Declaration of 1975, as revised in 2013.
Response:
Thank you for pointing this out. We agree with you. (Line 93-95.)
Q3:
Study participants- the creation of the datasets should be better described- how were patients collected.
MRI protocol- For dataset please state what complete MRI sequence (TR, TE, FOV, etc) were used for the MRI protocol.
Response:
Thank you very much for your suggestions, participant descriptions and the full MRI protocol have been revised in the original text. (Line 95-108.)
Q4:
The conclusion focuses solely on the technical aspects of the deep learning model and its performance in segmenting structures from MR images. It would be beneficial to expand the discussion to include the potential clinical impact of these findings and how they could contribute to improving patient outcomes or medical decision-making.
Agreed! “Currently, the three-dimensional visualization model of the lumbar spine plays a crucial role in preoperative planning and path trajectory evaluation for lumbar surgery [27, 28]. In this study, we have meticulously elucidated the anatomical structure adjacent to the lumbar intervertebral foramen and anterior to the lumbar spine, which will greatly facilitate preoperative planning of minimally invasive procedures in the lumbar region (including MIS oblique approach and intervertebral foramen endoscopy, among others). This may potentially reduce surgical complications and improve postoperative recovery outcomes.” the discussion has been put in the original article. (Line 338-345.)
27. Yang B, Fang SB, Li CS, Yin B, Wang L, Wan SY, Xie JK, Ding Q, Tang L, Zhong SZ. Digital three-dimensional model of lumbar region 4-5 and its adjacent structures based on a virtual Chinese human. Orthop Surg. 2013 May;5(2):130-4.
28. Jiang Y, Wang R, Chen C. Preoperative Simulation of the Trajectory for L5/S1 Percutaneous Endoscopic Transforaminal Discectomy: A Novel Approach for Decision-Making. World Neurosurg. 2021 Jan;145:77-82.
Reviewer 2 Report
The paper "Deep learning based automated magnetic resonance image segmentation of the lumbar structures and its adjacent structures at the L4/5 level" develop a deep learning model based on a 3D Deeplab V3+ network to automatically segment multiple structures from MRI at the L4/5 level. In this study, a total of 50 subjects were enrolled and 3D T2-SPACE images were acquired for the training, validation, and test analysis. The results showed that the modified 3D Deeplab V3+ network was helpful to automatically segment lumbar structures and their adjacent structures from MR images at the L4/5 level with an overall average DSC of 0.886, precision of 0.899, and recall of 0.881 on the test set. A strong correlation between manual and automatic segmentations was noted. Although the study showed some interesting results, I have some comments.
1) The case number was small (N=50). In deep learning, it usually requires
hundreds of case numbers to train a stable and reliable model. Thus, increasing number of subjects is needed for deep learning.
2) When the case number was insufficient, it is suggested to perform data augmentation for model training. However, in this study, no augmentation was performed under such a small number of subjects (N=50).
3) Although the study showed an overall average DSC of 0.886, the model was only tested on 50 (healthy?) subjects but was not tested on patients with abnormal structures.
4) The model was build using 3D T2-SPACE images. However, such a imaging dataset was not used in routine diagnosis. Can this model be reliable using 3D T1-weighted images.
5) Figure 3. Please define DAA, PI, DIVC. What are the red lines?
6) Figures 6 and 7. Figure resolution is too low.
Need to check for typos
Author Response
POINT-TO-POINT RESPONSE TO REVIEWERS
We would like to thank the reviewers and the Editors for their effort in evaluating this work and for their constructive comments and criticisms. The author team has discussed the review and carefully conducted a series of experiments to address the points raised. Several editorial changes were made in the main manuscript and those were indicated , as requested by the journal and uploaded separately. The outcome of the review process has been a stronger study and therefore we are grateful.
Reviewer(s)' Comments to Author:
Reviewing: 2
Review Comments for Author(s)
Q1:
The case number was small (N=50). In deep learning, it usually requires hundreds of case numbers to train a stable and reliable model. Thus, increasing number of subjects is needed for deep learning.
Response:
Thank you for your constructive suggestion. In the study, the case number was small but is feasible for image segmentation, which has been validated by the previous works1. In our study, to enhance generalization, the training dataset was augmented online with random rotations ranging from -15° to 15° and random elastic deformations.(Line 153-155) We hope to compensate for the lack of data volume through data augmentation.
1.Zheng G, Chu C, Belavý D L, et al. Evaluation and comparison of 3D intervertebral disc localization and segmentation methods for 3D T2 MR data: A grand challenge[J]. Medical image analysis, 2017, 35: 327-344.
Q2:
When the case number was insufficient, it is suggested to perform data augmentation for model training. However, in this study, no augmentation was performed under such a small number of subjects (N=50).
Response:
We sincerely thank the reviewer for careful reading. We have made extensive corrections to our previous draft. The detailed corrections are listed as follow: “ To enhance generalization, the training dataset was augmented online with random rotations ranging from -15° to 15° and random elastic deformations.”.(Line 153-155)
Q3:
Although the study showed an overall average DSC of 0.886, the model was only tested on 50 (healthy?) subjects but was not tested on patients with abnormal structures.
Response:
Thank you for pointing this out. The study enrolled 50 participants, including those with clinically predicted lumbar degenerative diseases as well as healthy volunteers. On 50 L4/5 levels, 41 were healthy and 9 unhealthy with spinal stenosis at 1 level, disc herniation at 8 levels, or both at 1 level. (Line 95-107)
Q4:
The model was build using 3D T2-SPACE images. However, such a imaging dataset was not used in routine diagnosis. Can this model be reliable using 3D T1-weighted images.
Response:
Thank you for pointing this out. The magnetic resonance images of the 3D-SPACE sequence have the advantages of high resolution, multi-plane reconstruction, and thin-layer scanning 1, 2. Its unique advantages include the imaging display of soft tissue structures, the convenience of observing the morphological changes of fine anatomical structures, and the advantages of multi-plane reconstruction 3, which are conducive to image segmentation by medical professionals or automatic recognition of image anatomical contours by computers, and automatic extraction of unique features of images. At present, our model has not been applied to conventional image sequences such as 3D T1 weighting. In the future, we plan to apply the model to conventional image sequences to enhance its universality.
- Baumert, K. Wörtler, D. Steffinger, G.P. Schmidt, M.F. Reiser, A. Baur-Melnyk, Assessment of the internal craniocervical ligaments with a new magnetic resonance imaging sequence: three-dimensional turbo spin echo with variable flip-angle distribution (SPACE), Magnetic resonance imaging 27(7) (2009) 954-960.
- Sayah, A.K. Jay, J.S. Toaff, E.V. Makariou, F. Berkowitz, Effectiveness of a rapid lumbar spine MRI protocol using 3D T2-weighted SPACE imaging versus a standard protocol for evaluation of degenerative changes of the lumbar spine, American Journal of Roentgenology 207(3) (2016) 614-620.
- J. Wang, Y. Wu, Z. Yao, Z. Yang, Assessment of pituitary micro-lesions using 3D sampling perfection with application-optimized contrasts using different flip-angle evolutions, Neuroradiology 56(12) (2014) 1047-1053.
Q5:
Figure 3. Please define DAA, PI, DIVC. What are the red lines?
Response:
Thank you for your reminder. The definitions of DAA, PI, and DIVC have been included in the original text(Lines 182-185).
Q6:
Figures 6 and 7. Figure resolution is too low.
Response:
Thank you for your reminder. We have improved the resolution of Fig. 6 and Fig. 7 in the original text
Reviewer 3 Report
The authors performed a very nice image segmentation study to assess the size of the surgical window for anterior approaches to the L4-5 interspace for spine surgery.
The exclusion criteria they used (eg spondylolistehsis), the young age and the low BMI of the patients limits the generalizability of the study as these values are different compared with the average surgical patient. This should be noted in the limitations.
Line 114: the term “celiac” is not widely used, I suggest changing it.
Line 122: the D is not present in the equation. Was D omitted?
Line 141-148: I am a bit confused. Was a true test dataset used? A test dataset should be seen by the algorithm only for evaluation and not for any kind of training/validation.
Line 166: what is the purpose of evaluating “a month” after the first analysis?
Figure 3: all abbreviations in the image need to have explanations in the figure legend.
Line 170: what is the OC abbreviation? What is O, what is C?
Table 2: the N4 performance was low, some explanation in the discussion is needed
Figure 5: the N4 performance is not excellent, I disagree with the figure legend.
The discussion needs some comments on why they chose DeepLabv3+, was it due to Atrous Convolution/Dilated Convolution?
What are the differences in the methodology compared with the previous studies mentioned in the discussion? Mentioning just some different performances might be due to different sizes of training sets. The authors need to establish what is the novelty of the current study compared with the two previous.
Author Response
POINT-TO-POINT RESPONSE TO REVIEWERS
We would like to thank the reviewers and the Editors for their effort in evaluating this work and for their constructive comments and criticisms. The author team has discussed the review and carefully conducted a series of experiments to address the points raised. Several editorial changes were made in the main manuscript and those were indicated , as requested by the journal and uploaded separately. The outcome of the review process has been a stronger study and therefore we are grateful.
Reviewer(s)' Comments to Author:
Reviewing: 3
Review Comments for Author(s)
Q1:
The authors performed a very nice image segmentation study to assess the size of the surgical window for anterior approaches to the L4-5 interspace for spine surgery.
The exclusion criteria they used (eg spondylolistehsis), the young age and the low BMI of the patients limits the generalizability of the study as these values are different compared with the average surgical patient. This should be noted in the limitations.
Response:
Thank you for your constructive suggestion. “Besides, the study cohort primarily consists of younger individuals, thus necessitating a larger sample size to encompass diverse age groups and body types in order to enhance the model's generalizability. ” We have added this detailed information in the newly revised manuscript.(Line 353-356)
Q2:
Line 114: the term “celiac” is not widely used, I suggest changing it.
Response:
Agreed! “the great celiac vessels” change as “the abdominal aorta, the inferior vena cava”. We have made changes in the original text.(Line 121-122)
Q3:
Line 122: the D is not present in the equation. Was D omitted?
Response:
Thank you for your reminder. “Given an image with specifications of I ∈RD∗H∗W.”. We have added it to the original text. (Line 126)
Q4:
Line 141-148: I am a bit confused. Was a true test dataset used? A test dataset should be seen by the algorithm only for evaluation and not for any kind of training/validation.
Response:
Thank you for your valuable question. Actually, we split the whole dataset as training dataset, validation dataset and test dataset in each fold. As described in the article, 10 subjects from a group were used as the test dataset in each experiment whereas 32 individuals, randomly selected from the other 4 groups (i.e., 40 subjects), were used as the training dataset, and the remaining 8 subjects were used for validation. Therefore, the test dataset was only used for testing the performance of the model and not for training or validation.
Q5:
Line 166: what is the purpose of evaluating “a month” after the first analysis?
Response:
Thank you for pointing this out. Repeating measurement data at intervals can increase the independence of the two measurements to a certain extent and reduce the subjectivity of repeated measurements; Thereby increasing the reliability of the data.
Q6:
Figure 3: all abbreviations in the image need to have explanations in the figure legend.
Response:
Thank you for your reminder. The definitions of DAA, PI, and DIVC have been included in the original text(Line 182-185).
Q6:
Line 170: what is the OC abbreviation? What is O, what is C?
Response:
Thank you for your reminder. OC is the abbreviation for ‘the oblique corridor’. We have made changes in the original text.(Line 324)
Q7:
Table 2: the N4 performance was low, some explanation in the discussion is needed
Response:
Thank you for pointing this out. “In addition, the intricate structure of nerve roots poses challenges in distinguishing them from other tissues. However, the overall segmentation performance of the model is satisfactory”We have added it to the original text.(Line 314-316)
Q8:
Figure 5: the N4 performance is not excellent, I disagree with the figure legend.
Response:
Thank you for pointing this out. We have changed ‘excellent’ to ‘well’ in the newly revised manuscript.(Line 222)
Q9:
The discussion needs some comments on why they chose DeepLabv3+, was it due to Atrous Convolution/Dilated Convolution?
Response:
Thank you for your constructive suggestion. We discussed the previous draft. Specific discussions are as follows: “The modified 3D Deeplabv3+ network has been well demonstrated in this study due to its multi-scale and multi-structural segmentation advantages”. (Line 275-277). At the same time, atrous convolution/filtered convolution1,2 can reduce the number of model parameters and improve generalization ability.
- Wang P , Chen P , Yuan Y ,et al.Understanding Convolution for Semantic Segmentation[C]//2018 IEEE Winter Conference on Applications of Computer Vision (WACV).IEEE, 2018.
- Chen LC, Papandreou G, Kokkinos I, Murphy K, Yuille AL. DeepLab: Semantic Image Segmentation with Deep Convolutional Nets, Atrous Convolution, and Fully Connected CRFs. IEEE Trans Pattern Anal Mach Intell. 2018 Apr;40(4):834-848.
Q10:
What are the differences in the methodology compared with the previous studies mentioned in the discussion? Mentioning just some different performances might be due to different sizes of training sets. The authors need to establish what is the novelty of the current study compared with the two previous.
Response:
Thank you for your constructive suggestion. “In contrast to previous research, this study has segmented multiple structures in the lumbar spine and its vicinity, including bones, intervertebral discs, nerve roots, dura mater, abdominal aorta, inferior vena cava and psoas major. This provides a more detailed three-dimensional anatomical relationship that significantly enhances the clinical applicability of the segmentation model.” We have added this detailed information in the newly revised manuscript.(Line 300-304)
Round 2
Reviewer 1 Report
The authors have satisfactorily addressed my concerns.
Author Response
Reviewer(s)' Comments to Author:
Reviewing: 1
Review Comments for Author(s)
Q1:
The authors have satisfactorily addressed my concerns.
Response:
We appreciate that reviewer #1 has been satisfied with this revised version.
Reviewer 2 Report
Lines 114-116. Why were those subjects with spinal deformity and lumbar spondylolisthesis excluded from the study? If the proposed model cannot be utilized in those with such diseases, it should be addressed in the limitation.
Minor English check is needed.
Author Response
Reviewing: 2
Review Comments for Author(s)
Q1:
Lines 114-116. Why were those subjects with spinal deformity and lumbar spondylolisthesis excluded from the study?
Response:
We sincerely thank the reviewer for careful reading. Patients with lumbar vertebral deformities may exhibit incomplete vertebral anatomy, while those with lumbar spondylolisthesis often present with fractures in the vertebral isthmus, which can compromise the integrity and continuity of the vertebral anatomical structure and potentially impact segmentation model research. Therefore, this study excludes patients with lumbar spine deformities and spondylolisthesis as part of its exclusion criteria. Furthermore, in clinical practice, the inclusion of patients with lumbar spine deformities and spondylolisthesis would be a captivating avenue for future research, warranting our focused attention in subsequent investigations.
Q2:
If the proposed model cannot be utilized in those with such diseases, it should be addressed in the limitation.
Response:
Thank you for your constructive suggestion. “Furthermore, the study cohort primarily consisted of younger individuals with relatively uncomplicated disease types, predominantly lumbar disc herniation and spinal stenosis patients. Therefore, we intend to expand our sample size to encompass diverse age groups, body types, and disease types (including spinal deformity and lumbar spondylolisthesis) to enhance the generalizability of our model.”We have added it to the original text. (Line 353-358)
